# Dynamics of the State of Arterial Stiffness as a Possible Pathophysiological Factor of Unfavorable Long-Term Prognosis in Patients after Coronary Artery Bypass Grafting

**DOI:** 10.3390/biomedicines12051018

**Published:** 2024-05-06

**Authors:** Alexey N. Sumin, Anna V. Shcheglova, Olga L. Barbarash

**Affiliations:** Federal State Budgetary Institution “Research Institute for Complex Issues of Cardiovascular Disease”, Blvd. Named Academician L.S. Barbarasha, 6, 650002 Kemerovo, Russia; an_sumin@mail.ru (A.N.S.); olb61@mail.ru (O.L.B.)

**Keywords:** cardio-ankle vascular index, dynamics, coronary artery disease, coronary artery bypass grafting, long-term prognosis

## Abstract

The aim of this study was to examine the long-term prognostic value of changes in the cardio-ankle vascular index (CAVI) within a year after coronary artery bypass grafting (CABG). Methods. Patients with coronary artery disease (*n* = 251) in whom CAVI was assessed using the VaSera VS-1000 device before and one year after CABG. Groups with improved CAVI or worsened CAVI were identified. We assessed the following events at follow-up: all-causes death, myocardial infarction, and stroke/transient ischemic attack. Results. All-causes death was significantly more common in the group with worsened CAVI (27.6%) than in the group with CAVI improvement (14.8%; *p* = 0.029). Patients with worsened CAVI were more likely to have MACE, accounting for 42.2% cases, compared with patients with CAVI improvement, who accounted for 24.5%; *p* = 0.008. Worsened CAVI (*p* = 0.024), number of shunts (*p* = 0.006), and the presence of carotid stenosis (*p* = 0.051) were independent predictors of death from all causes at 10-year follow-up after CABG. The presence of carotid stenosis (*p* = 0.002) and the group with worsened CAVI after a year (*p* = 0.008) were independent predictors of the development of the combined endpoint during long-term follow-up. Conclusions. Patients with worsening CAVI one year after CABG have a poorer prognosis at long-term follow-up than patients with improved CAVI. Future research would be useful to identify the most effective interventions to improve CAVI and correspondingly improve prognosis.

## 1. Introduction

Long-term prognosis in patients with coronary artery disease (CAD) after coronary artery bypass grafting (CABG) depends not only on preoperative factors (initial severity of coronary disease and myocardial condition, comorbidity, level of pathological biomarkers) [1,2,3], but also from further treatment and rehabilitation measures. The task of secondary prevention in this category of patients is to level out those unfavorable risk factors that led to the CAD development, and, ultimately, to the need for myocardial revascularization [4,5]. Not surprisingly, most studies have focused on identifying baseline factors associated with long-term prognosis after CABG. For example, recent studies have shown the prognostic impact of biomarkers such as lipoprotein (a) [3] and LDL/HDL ratio [2] during 10-year follow-up of patients after CABG. The presence of a large number of risk factors and biomarkers that need to be adjusted and monitored when managing this category of patients creates a certain inconvenience for practitioners and can lead to excessive expenditure of resources.

Therefore, it seems attractive to use such an integral indicator that reflects the influence of various risk factors, as well as arterial wall stiffness. Classically, it is determined by assessing the pulse wave velocity [6]; however, this method has certain limitations (dependence on blood pressure level, inconvenience of performing the study, dependence on the qualifications of personnel) [6], so another indicator has been proposed—the cardio-ankle vascular index (CAVI) [7]. This indicator is based on determining the rigidity parameter β, which reflects the degree of dependence of pressure on volume; therefore, the CAVI index does not depend on the level of blood pressure. This makes it potentially suitable for studying the state of vascular stiffness over time [8].

Currently, prospective epidemiological studies have shown that CAVI is associated with the development of cardiovascular events [9,10]. Moreover, the presence of pathological CAVI is associated with an unfavorable prognosis in patients with various forms of coronary artery disease: with acute coronary syndrome [11,12], with stable CAD [13], after CABG [14,15]. In our previous study, we showed that CAD patients with abnormal CAVI before CABG experienced more frequent adverse events and death during long-term follow-up than patients with normal CAVI. It was also found that the presence of subclinical multifocal atherosclerosis and pathological CAVI were independent predictors of the development of the combined endpoint [15]. Since in this cohort of patients, we have the results of assessing CAVI not only before CABG, but also one year after surgery, it became possible to assess the impact of not only one-time, but also serial assessment of CAVI on the prognosis. However, the possible influence of the dynamics of CAVI on the long-term prognosis in this category of patients remains unclear. Since there are data on the impact of treatment interventions on CAVI values [16,17], a logical question arises: does a change in this index (or lack of change) affect the prognosis? To date, only a few studies [13,18] have been conducted in this direction. This motivated the present study, which aims to examine the long-term prognostic value of changes in CAVI within a year after CABG. 

## 2. Material and Methods

### 2.1. Study Population

Initially, the study included 732 consecutive patients (age from 33 to 81 years) in the cardiology department of the Federal State Budgetary Institution “Research Institute of Complex Problems of Cardiovascular Diseases” (Kemerovo) for planned coronary bypass surgery (2012–2013). A cohort of 545 patients was recruited and underwent arterial stiffness testing with cardio-ankle vascular index (CAVI) using the VaSera automated device (Fukuda Denshi, Tokyo, Japan). The criteria for inclusion and exclusion of patients from the study were presented in detail in previously published articles [14,15]. Thus, the initial analyzed sample consisted of 356 patients. At this stage, 125 patients (35.1%) had CAVI values of 9.0 or more. Before the study, written informed consent was obtained from all patients. This study was approved by the local ethics committee of the institution, and the study protocol complied with the ethical principles of the Declaration of Helsinki.

One year after surgery, patients were contacted by telephone and their presence at the study center was ensured. One year after surgery, 251 patients (71%) were able to remeasure CAVI, and we tracked changes and classified them as having improved CAVI or worsened CAVI. The improved CAVI group included patients with a decrease in CAVI from a pathological value (CAVI ≥ 9.0) to normal (CAVI < 9.0), or the index remained within normal values. The worsened CAVI group included patients with a persistent pathological index value (CAVI ≥ 9.0) or an increase from CAVI < 9.0 to CAVI ≥ 9.0 or an increase of 1 unit or more). If the CAVI value was ≥ 9.0 on at least one side, the index was considered pathological. All patients were prescribed optimal drug therapy and given recommendations for lifestyle changes in accordance with the recommendations.

### 2.2. Evaluation of Indicators

For patients with coronary artery disease, clinical and laboratory data were assessed before surgery and one year after it. The examination consisted of determining serum cholesterol and low-density lipoprotein cholesterol (LDL-C), fasting glucose, ultrasound examination of the heart and carotid arteries, and coronary angiography in accordance with the protocol specified in our previous articles [14].

CAVI measurements were carried out using the VaSera VS-1000 device (Fukuda Denshi, Tokyo, Japan) in a quiet room in the morning in a supine position. The index was calculated automatically on the right and left. Four occlusive cuffs were applied to the shoulders and legs on the right and left, ECG electrodes were placed on the wrists, and a PCG microphone was placed in the second intercostal space to the left of the sternum edge to obtain a PCG signal. The index is calculated automatically by the device on the right and left lower extremities [8].

### 2.3. Follow-Up

After surgery, patients were followed up for 9.7 ± 0.9 years. Data were collected through active telephone monitoring and medical information system analysis. Long-term information was collected on 210 (83.7%) patients. In the long-term period, data on the condition of patients, drug therapy were analyzed and hard endpoints were recorded, such as coronary and non-coronary death, non-fatal myocardial infarction (MI) and acute cerebrovascular accident (Stroke/TIA). Among 210 patients, two groups were formed: the first included 94 (44.8%) patients with improved CAVI one year after coronary artery bypass grafting, the second group included 116 (55.2%) patients whose CAVI status worsened one year after CABG. The flow diagram of the study is detailed in Figure 1.

### 2.4. Statistical Analyses

For statistical processing, the programs “STATISTICA 8.0” (Dell Software, Inc., Round Rock, TX, USA) and SPSS 17.0 (IBM, Armonk, NY, USA) were used. To decide on the distribution of quantitative variables, the Kolmogorov–Smirnov test was used. To present quantitative variables with a distribution other than normal, the median and quartiles (lower and upper) were used. To compare two independent groups based on quantitative characteristics, the Mann–Whitney test was used. Qualitative values were presented in absolute numbers (*n*) and percentages (%), and comparisons between the groups were performed using χ^2^ tests. Kaplan–Meier curves were used to estimate the long-term survival rates and long-term event-free survival rates in the two groups (with improved CAVI and with worsened CAVI). Differences in survival rates between groups were analyzed with log-rank tests. The binary logistic regression analysis (Forward LR method) was used to assess the relationship of binary signs (all-cause mortality; combined end point—all-cause death + non-fatal myocardial infarction + non-fatal stroke) with preoperative indicators, with the data of arterial stiffness after a year, and with dynamics CAVI at one year. Performance of arterial stiffness parameters in recognizing the risk of unfavorable prognosis (all-cause death, development of a composite endpoint) after CABG was evaluated through receiver operating characteristic curve analysis. The level of statistical significance was defined as *p* < 0.05.

## 3. Results

Table 1 presents baseline characteristics comparing clinical parameters of patients with worsening/persistent abnormal CAVI and improved/persistent normal CAVI. There were no statistically significant differences in clinical variables between groups, with the exception of a history of diabetes mellitus (*p* = 0.011).

The dynamics of the clinical manifestations of coronary and heart failure, as well as the laboratory and instrumental characteristics of patients, at the time of inclusion in the study and one year after surgery, are presented in Table 2. In the group of patients with CAVI progression, clinical manifestations of stage II-III CHF initially prevailed NYHA (*p* = 0.009). However, one year after surgery, the clinical manifestations of CHF decreased in both groups and there were no statistically significant differences between the groups.

At baseline, total cholesterol and LDL-C levels did not differ, but one year after CABG, total cholesterol and LDL-C levels were higher in the CAVI improved group, and the differences were statistically significant (*p* < 0.05). Echocardiographic parameters, including left ventricular ejection fraction and E/A ratio, did not differ between groups. The dynamics of CAVI and ABI are also reflected in Table 2. In an intergroup comparison, the initial values of the indices on the right and left were comparable in both groups; after a year, the level of CAVI was naturally higher in the group with its progression (*p* < 0.001).

As a screening procedure, ultrasound examination of the carotid arteries was performed initially and over time. Stenosis of the carotid arteries of 30% or more was detected equally often in both groups in 18.1% of cases; however, after a year, progression of atherosclerosis of the carotid arteries ≥ 30% was noted for both groups (with an improvement in CAVI 24.4%, with a worsening CAVI 23.7%, *p* > 0.05). When analyzing the initial anatomical characteristics of the coronary arteries, no statistically significant differences were found between the groups (Table 3), which also explains the lack of differences in the number of coronary bypass grafts applied during surgery (Table 4). The groups were comparable in terms of the total duration of the operation and the duration of artificial circulation. In both groups, combined operations with CABG were performed equally often.

The reception of optimal drug therapy (OMT) (angiotensin converting enzyme inhibitors, angiotensin receptor blockers, beta-blockers, calcium channel blockers, statins, antiplatelet agents) during the entire observation period is presented in Table 5. In the prehospital period, the frequency of OMT use was low and comparable in both groups. During their hospital stay, all patients received standard therapy. After a year, there was a trend towards a higher frequency of taking OMT, without intergroup differences; however, in the long-term period, low adherence to the prescribed treatment was noted in all patients and the groups did not differ statistically significantly.

During the long-term follow-up period, 75 (35.7%) cardiovascular events were recorded, including all cause death in 46 (21.9%) patients, non-fatal myocardial infarction in 12 (5.7%), and stroke in 17 (8.1%) patients.

Death from all causes was significantly more common in the group with CAVI progression—32 (27.6%), than in the group with CAVI improvement—14 (14.8%; *p* = 0.029). Death from cardiac causes was also more common in the worsening group than in the improving CAVI group (14.7% vs. 9.6%, *p* = 0.266). Patients with CAVI progression were more likely to have MACE (all cause death, MI, stroke/TIA), with 49 (42.2%) cases, compared to patients with CAVI improvement, who had 23 (24.5%) cases, *p* = 0.008 (Figure 2, Appendix A).

The Kaplan–Meier curves uncovered a worse long-term prognosis in the group with worsened CAVI relative to the group with improved CAVI (Figure 3 and Figure 4). The differences were significant for event-free survival rates (Figure 4, Appendix A; *p* = 0.034, *p* = 0.018, and *p* = 0.023, respectively, for log-rank, Breslow, and Tarone-Ware tests). But, the differences were not significant for survival rates (Figure 3, Appendix A; *p* = 0.109, *p* = 0.070, and *p* = 0.081, respectively, for log-rank, Breslow, and Tarone-Ware tests).

In the multiple binary logistic regression model (direct LR method), the following factors had a significant relationship (χ^2^(3) = 16.671, *p* = 0.001) with death from all causes during long-term follow-up after CABG: group with worsened CAVI (B = 0.852, *p* = 0.024), number of shunts (B = 0.627, *p* = 0.006), and the presence of carotid stenosis more than 30% (B = 0.843, *p* = 0.051). This model explained only 12.7% (Nagelkerke R^2^) of the variance in all-cause death and correctly classified 77.6% of cases (Table 6). The presence of carotid stenosis more than 30% (B = 1.208, *p* = 0.002) and the group with worsened CAVI after a year (B = 0.853, *p* = 0.008) were independent predictors of the development of the combined endpoint during long-term follow-up after CABG. For this model, the statistical significance was χ^2^(2) = 16.736, *p* = 0.001, the Nagelkerke R^2^ value was 0.115, and the model correctly classified 67.2% of cases (Table 6).

Among the indicators of arterial stiffness (R_CAVI and L_CAVI basal, R_CAVI and L_CAVI after a year, dynamics of CAVI after a year), the greatest association with the development of death from all causes during long-term follow-up was noted for the group with worsened CAVI after a year (Table 7 and Figure 5). Similar data were obtained for the development of the combined endpoint during long-term follow-up (Table 8 and Figure 6). However, the curve area of this variable was <0.7 in both cases, indicating unacceptable discrimination.

## 4. Discussion

In the present study, one year after CABG surgery, almost half of the patients showed positive dynamics or stable normal CAVI values. At ten-year follow-up, these patients, when compared with patients with worsened or persistently pathological CAVI, showed a decrease in all-cause mortality and the incidence of the composite endpoint. Among the independent predictors of overall mortality and development of a composite endpoint were worsened CAVI during the year and the presence of subclinical carotid stenoses during the preoperative examination.

One of the advantages of CAVI over other indicators of arterial stiffness is its independence from blood pressure levels and, accordingly, the possibility of dynamic assessment. Despite this, the study of prognostic assessment of such dynamics is still infrequently used. Thus, in patients with dyslipidemia and risk factors, unfavorable dynamics of CAVI during the first year positively correlated with the development of MACE at five-year follow-up. At the same time, Cox proportional hazards regression analysis failed to show an independent effect of CAVI on the risk of developing MACE [18]. Apparently, the initial cohort of subjects (without an established CAD diagnosis) did not allow us to identify such an effect due to the low frequency of MACE in it. To date, only one study has examined the effect of CAVI dynamics on prognosis in patients with CAD. Otsuka T et al. [13] showed that in patients with newly CAD diagnosed, when assessing CAVI after 6 months, this index improved in only half of the patients; in the rest, it remained persistently elevated. At 3-year follow-up, persistently elevated CAVI was an independent predictor of future MACE. In our study, the frequency of positive changes in CAVI over the course of a year was comparable, and we were also able to confirm the association of changes in CAVI with prognosis over a longer period of observation in a cohort of patients after CABG. Although in a previous study we showed the negative impact of the pathological CAVI index before CABG on the 10-year postoperative prognosis [15], in the present study, the persistence of pathological CAVI during the year turned out to have a more pronounced prognostic significance.

Another option for studying the association of the dynamics of CAVI with prognosis is to study it against the background of psycho-emotional stress. Dynamic observation showed that in response to severe stress (earthquake in Japan), there was a significant increase in CAVI and, as a consequence, the number of cardiovascular events in patients with coronary artery disease [19]. Also, the association between stress level and arterial stiffness is evidenced by a study by Tajima T. et al. [20]. In a study of healthy individuals, they showed that in women, an increase in salivary alpha-amylase activity (a biomarker for chronic psychological stress) was associated with an increase in the CAVI index. Significantly, in healthy young adults, even a 5 min mental counting test resulted in a significant increase in arterial stiffness within 30 min [21] Interesting observations are presented in the article by Shimizu K et al. [22]—immediately before MACE (myocardial infarction, cerebral hemorrhage, and aortic dissection), patients tended to rapidly increase in CAVI from ~0.5 to 1.0 over the several weeks immediately preceding the event. The above scientific and clinical data have led to the advancement of the “smooth muscle cell contraction hypothesis” as the cause of plaque rupture. The authors of this hypothesis proposed that MACEs occur due to plaque rupture due to ischemic injury and necrosis caused by the rapid increase in CAVI in the background of an initially elevated CAVI [23]. It can also be assumed that increased arterial stiffness is a factor mediating the effect of psychoemotional stress on prognosis. In addition, it is the initially increased arterial stiffness that contributes to the implementation of psychoemotional stress as a trigger for the development of MACE in cardiac patients.

The clinical significance of this study seems to us to be versatile. First, it emphasizes the need to assess CAVI over time to identify the group of patients at greatest risk of developing MACE in the future. Secondly, it seems appropriate to use CAVI as an indicator of the effectiveness of therapeutic and preventive measures. Herewith, the range of influences can be very wide—from educational activities [24], physical training [25], and lifestyle correction [16], to the appointment of optimal drug therapy [26]. At the same time, assessing the dynamics of CAVI may be useful in assessing the effectiveness of therapy, since even drugs of the same group can have different effects on arterial stiffness [17].

Considering the adverse effect of stress on arterial stiffness, which mediates the development of MACE, another direction in the treatment of such patients may be behavioral therapy. Currently, there are no studies examining the effect of stress-limiting therapy on improving vascular function, as already noted [27]. Accordingly, future research is needed to determine whether stress-reducing behavioral interventions can lead to reductions in cardiovascular events through improvements in CAVI.

### Study Limitations

When evaluating the results of this study, its limitations should be considered. First, we did not include patients with certain comorbid conditions (atrial fibrillation, low ejection fraction, ABI values less than 0.9, presence of valvular lesions) in order to be able to correctly evaluate CAVI. Thus, the predictive value of CAVI dynamics can only be attributed to this sample of patients and cannot be extended to all patients after CABG. Accordingly, our analysis did not include patients with peripheral arterial disease, which could influence the relative contribution of the presence of subclinical carotid stenosis and unfavorable changes in the CAVI index to long-term prognosis after CABG. Second, the sample size is relatively small, so our results can be considered preliminary, which should be confirmed in a study with a larger sample or in a multicenter study. Currently, such a study is already underway—the Cardiovascular Prognostic COUPLING Study [28], which will resolve the issue of confirming our data. Third, we did not conduct targeted monitoring of the therapy of the patients included in the study; they received treatment from doctors at their place of residence in accordance with current recommendations. However, inclusion of therapy received in multiple regression models did not reveal an independent effect on prognosis. It should also be noted that the AUC in the ROC is low, indicating an unacceptable ability of the studied factors to distinguish the development of adverse outcomes. In addition, it should be taken into account that the present study examined the same cohort of patients as in the authors’ previous work [15]. However, since that publication presented the results of a single assessment of CAVI (before CABG), and this article presents the prognostic value of a serial assessment of this index, we consider this publication important for further research in this area.

## 5. Conclusions

In the present study, one year after CABG surgery, 45% of patients showed positive dynamics or stable normal CAVI values. At ten-year follow-up, these patients, when compared with patients with worsened or persistently pathological CAVI, showed a decrease in all-cause mortality and the incidence of the composite endpoint. Among the independent predictors of overall mortality and development of combined endpoint were worsened CAVI during the year and the presence of subclinical carotid stenoses during preoperative examination. In further studies, it is necessary to study which interventions in patients after CABG can cause favorable dynamics of CAVI, to what extent such dynamics can improve the prognosis, and also whether behavioral interventions can improve CAVI or help reduce the development of MACE in such patients.

## Figures and Tables

**Figure 1 biomedicines-12-01018-f001:**
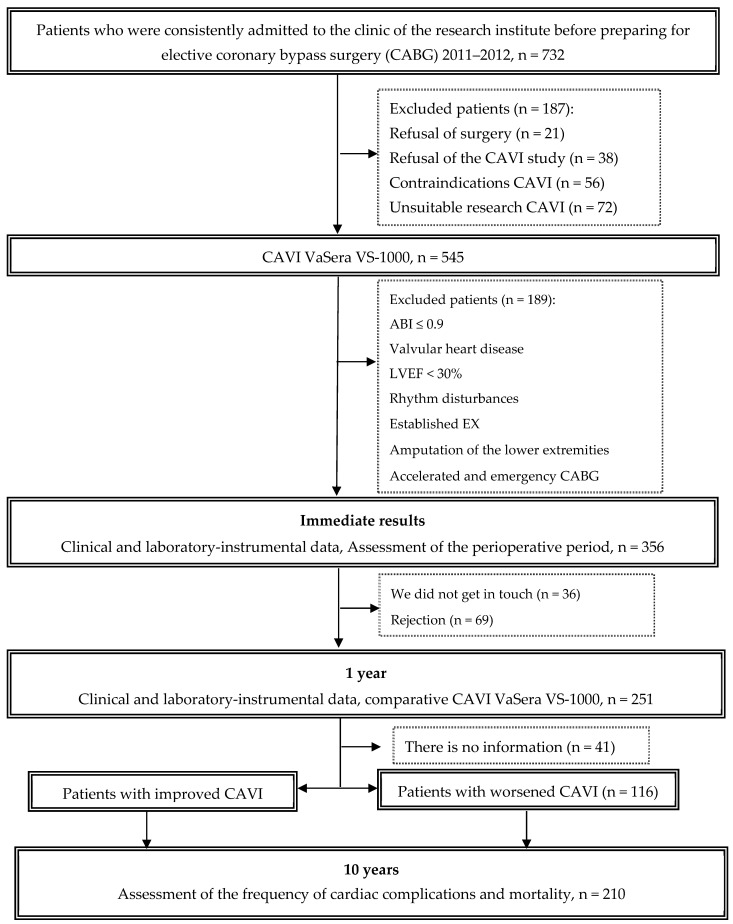
Flow chart of the study design from screening to completion of the trial. CAVI—cardio-ankle vascular index; ABI—ankle-brachial index; LVEF—left ventricular ejection fraction; CABG—coronary artery bypass graft.

**Figure 2 biomedicines-12-01018-f002:**
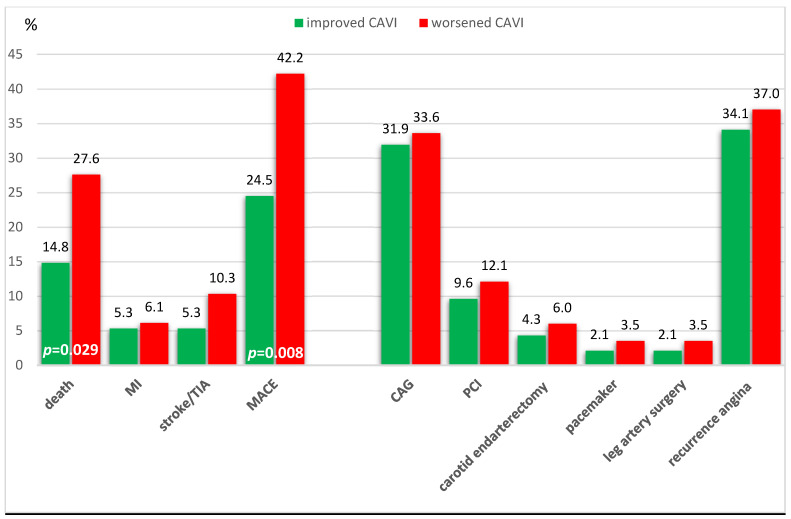
Complications of the ten-year period depending on the dynamics of CAVI in patients with coronary heart disease who underwent CABG. Abbreviations: CAG—Coronarography; PCI—percutaneous coronary intervention, TIA—transitory ischemic attack, MI—myocardial infarction, MACE—major adverse cardiac events.

**Figure 3 biomedicines-12-01018-f003:**
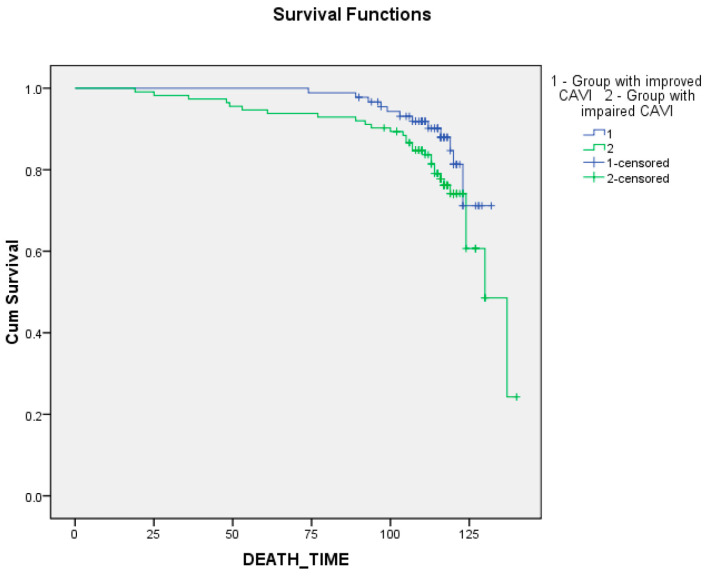
Impact CAVI changes within a year after coronary artery bypass grafting on long-term survival.

**Figure 4 biomedicines-12-01018-f004:**
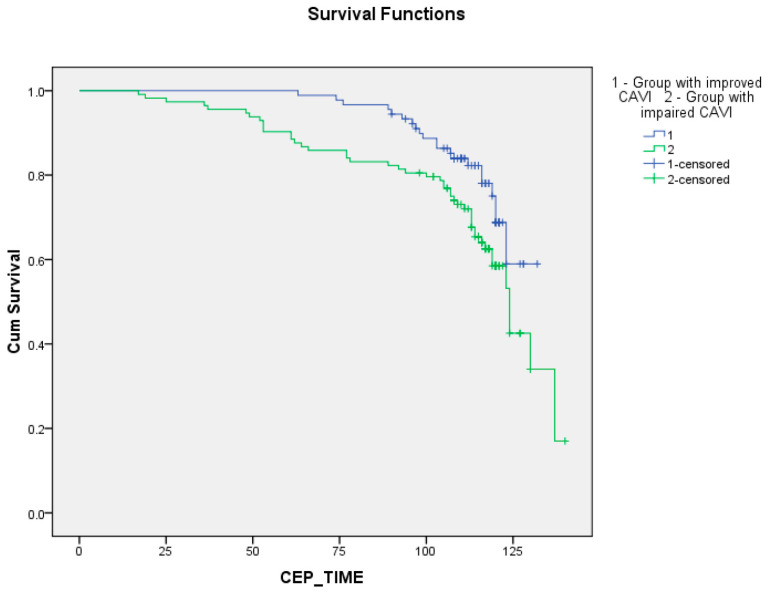
Impact CAVI changes within a year after coronary artery bypass grafting on long-term event-free survival.

**Figure 5 biomedicines-12-01018-f005:**
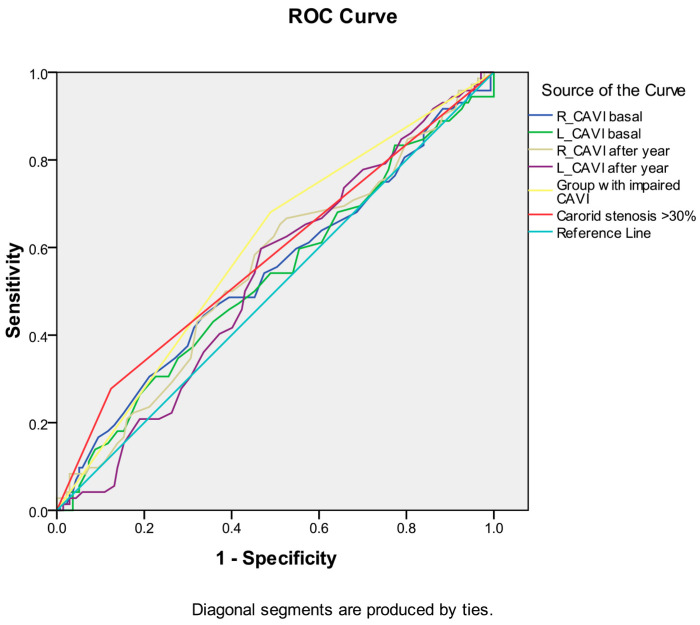
Receiver operating characteristic curve analysis. Performance of baseline parameters (values of CAVI and presence of carotid stenoses ≥ 30%) in discriminating combined endpoint development in ten-year period after CABG.

**Figure 6 biomedicines-12-01018-f006:**
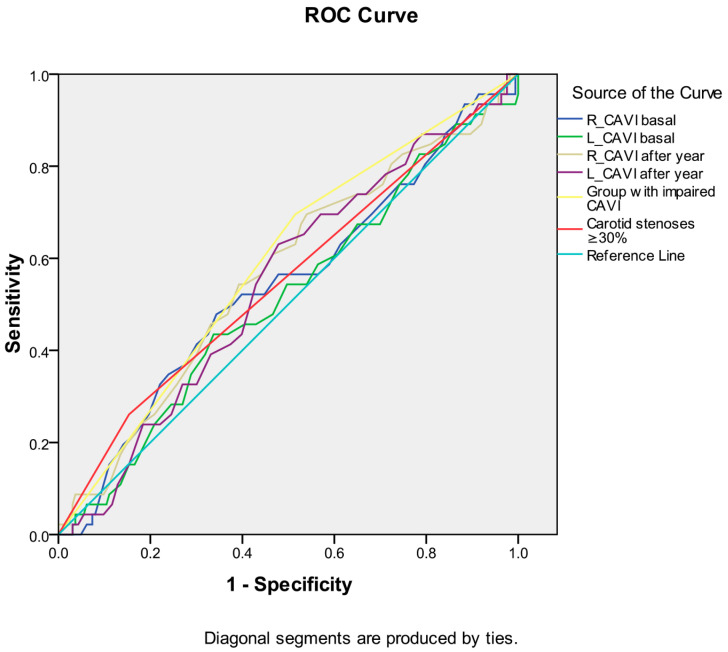
Receiver operating characteristic curve analysis. Performance of baseline parameters (values of CAVI and presence of carotid stenoses ≥ 30%) in discriminating all-cause mortality development in ten-year period after CABG.

**Table 1 biomedicines-12-01018-t001:** Comparison of the initial characteristics in groups with improved or worsened CAVI one year after CABG.

	Group with Improved CAVI (*n* = 94)	Group with Worsened CAVI (*n* = 116)	*p*-Value
Age, years ME [LQ; UQ]	57.5 [53.0; 64.0]	59.0 [55.0; 64.5]	0.118
Male, *n* (%)	66 (70.2)	87 (75.0)	0.437
Height, cm ME [LQ; UQ]	169.0 [163.0; 176.0]	170.0 [164.0; 176.0]	0.646
Weight, kg ME [LQ; UQ]	80.0 [72.0; 90.0]	80.0 [70.0; 89.0]	0.914
Body mass index, kg m^−2^ ME [LQ; UQ]	28.1 [24.6; 31.6]	28.0 [25.5; 31.1]	0.861
Myocardial infarction, (*n* %)	54 (57.5)	72 (62.1)	0.496
Hypertension, *n* (%)	73 (77.7)	104 (89.7)	0.017
Stroke, (*n* %)	6 (6.4)	9 (7.8)	0.7
Transischemic attack, (*n* %)	0	2 (1.72)	0.2
Diabetes, *n* (%)	7 (7.5)	23 (19.8)	0.011
PCI, (*n* %)	4 (4.3)	9 (7.8)	0.294
CABG, (*n* %)	1 (1.1)	1 (0.9)	0.881
Carotid endarterectomy, (*n* %)	0	3 (2.6)	0.116
Smoking experience, years ME [LQ; UQ]	30.0 [20.0; 40.0]	30.0 [20.0; 40.0]	0.478
Smoking, (*n* %)	20 (21.3)	32 (27.6)	0.292

Abbreviations: ME [LQ; UQ]—median with upper and lower quartile; CABG—coronary artery bypass graft; PCI—percutaneous coronary intervention.

**Table 2 biomedicines-12-01018-t002:** Comparison of clinical and laboratory-instrumental characteristics of data in groups with improved or worsened CAVI one year after CABG.

	Group with Improved CAVI (*n* = 94)	Group with Worsened CAVI (*n* = 116)	*p*-Value
The clinical characteristics			
Angina baseline (*n* %)	73 (77.7)	94 (81.0)	0.546
Angina one year follow-up (*n* %)	9 (9.6)	10 (8.6)	0.812
Heart failure II-III FC NYHA baseline (*n*, %)	18 (19.2)	41 (35.3)	0.009
Heart failure II-III FC NYHA one year follow-up (*n*, %)	12 (12.9)	15 (13.2)	0.956
Laboratory data			
Total cholesterol baseline, mmol/L	5.0 [4.2; 5.9]	4.6 [4.0; 5.6]	0.056
Total cholesterol one year follow-up, mmol/L	4.9 [4.1; 6.2]	4.35 [3.7; 5.5]	0.019
LDL cholesterol baseline, mmol/L	3.04 [2.1; 3.9]	2.8 [2.22; 3.5]	0.139
LDL cholesterol one year follow-up, mmol/L	3.2 [2.3; 3.8]	2.4 [1.9; 3.4]	0.017
Fasting glucose baseline, mmol/L basal	5.5 [5.0; 6.2]	5.6 [5.2; 6.5]	0.155
Fasting glucose one year follow-up, mmol/L	5.5 [5.0; 6.2]	5.8 [5.3; 6.9]	0.066
GFR CKD-EPI baseline, mL/min/1.73 m^2^	80.9 [65.5; 102.7]	82.1 [65.4; 97.7]	0.966
GFR CKD-EPI one year follow-up, mL/min/1.73 m^2^	92.8 [70.6; 112.7]	86.2 [71.4;106.0]	0.553
Echocardiography			
LV ejection fraction baseline, %	60.0 [54.0; 64.0]	60.0 [53.0; 64.0]	0.483
LV ejection fraction one year follow-up, %	62.0 [57.0; 65.0]	61.0 [51.0; 65.0]	0.116
E/A baseline	0.88 [0.7; 1.2]	0.8 [0.7; 1.1]	0.653
E/A one year follow-up	0.8 [0.51; 1.1]	0.68 [0.5; 1.0]	0.133
Vasera-1000			
R-CAVI baseline	8.4 [7.6; 9.3]	8.5 [7.9; 9.3]	0.458
R-CAVI one year follow-up	7.7 [7.1; 8.2]	9.1 [8.6; 9.7]	<0.001
L-CAVI baseline	7.6 [7.1; 8.1]	8.5 [7.9; 9.4]	0.371
L-CAVI one year follow-up	7.7 [7.1; 8.1]	9.1 [8.3; 9.6]	<0.001
R-ABI baseline	1.12 [1.03; 1.22]	1.14 [1.08; 1.2]	0.351
R-ABI one year follow-up	1.08 [099; 1.17]	1.09 [0.99; 1.19]	0.501
L-ABI baseline	1.1 [1.02; 1.16]	1.11 [1.04; 1.18]	0.402
L-ABI one year follow-up	1.03 [0.92; 1.11]	1.05 [094; 1.13]	0.117

Notes: FC—functional class; NYHA—New York Heart Association; LV—left ventricular; E/A—the ratio of the peak of the early to late transmitral flow; GFR—glomerular filtration rate, CKD-EPI—Chronic Kidney Disease Epidemiology Collaboration, LDL—low-density lipoprotein; CAVI—cardio-ankle vascular index; ABI—ankle-brachial index.

**Table 3 biomedicines-12-01018-t003:** Severity of damage to coronary arteries, non-coronary atherosclerosis in groups with improved or worsened CAVI one year after CABG.

	Group with Improved CAVI (*n* = 94)	Group with Worsened CAVI (*n* = 116)	*p*-Value
Coronary angiography			
LCA ≥ 50%, (*n* %)	17 (18.1)	29 (25.0)	0.228
One vessel disease coronary artery ≥ 70%, (*n* %)	17 (18.1)	26 (22.41)	0.429
Two vessel disease coronary artery ≥ 70%, (*n*, %)	30 (31.9)	38 (32.8)	0.896
Three vessel disease coronary artery (*n*, %)	40 (42.6)	46 (39.7)	0.671
Non-coronary atherosclerosis			
Carotid artery stenosis ≥ 30%, baseline (*n*, %)	17 (18.1)	21 (18.1)	0.997
Carotid artery stenosis ≥ 30%, one year follow-up (*n*, %)	22 (24.4)	27 (23.7)	0.899
Carotid artery stenosis ≥ 50%, baseline (*n*, %)	11 (11.7)	16 (13.8)	0.652
Carotid artery stenosis ≥ 50%, one year follow-up (*n*, %)	7 (7.8)	12 (10.5)	0.502

Notes: LCA—left coronary artery.

**Table 4 biomedicines-12-01018-t004:** The main characteristics of coronary artery bypass surgery in groups with improved or worsened CAVI one year after CABG.

	Group with Improved CAVI (*n* = 94)	Group with Worsened CAVI (*n* = 116)	*p*-Value
Euroscore (scores)	2.5 [1.0; 4.0]	2.0 [2.0; 4.0]	0.630
Euroscore, %	1.33 (0.88; 2.35)	1.54 (1.0; 2.4)	0.464
Cardiopulmonary bypass, (*n* %)	84 (89.4)	96 (82.8)	0.174
Number of shunts	3.0 [2.0; 3.0]	3.0 [2.0; 3.0]	0.542
Cardiopulmonary bypass time, min	98.5 [77.5; 110.0]	94.0 [84.0; 107.0]	0.795
Total operation time, min	240.0 [204.0; 300.0]	240.0 [195.0; 273.0]	0.330
Ventriculoplasty, (*n* %)	5 (5.32)	4 (3.5)	0.505
Thrombectomy, (*n*, %)	4 (4.3)	1 (0.9)	0.108
Carotid endarterectomy, (*n* %)	1 (1.1)	3 (2.6)	0.422
Radiofrequency ablation, (*n* %)	1 (1.1)	3 (2.6)	0.422

**Table 5 biomedicines-12-01018-t005:** Comparison of medication use in groups with improved or worsened CAVI one year after CABG.

	Group with Improved CAVI (*n* = 94)	Group with Worsened CAVI (*n* = 116)	*p*-Value
Preoperative medical treatment			
Beta-blockers, *n* (%)	62 (66.0)	74 (63.8)	0.943
CCBs, *n* (%)	20 (21.3)	25 (21.6)	0.883
Statins, *n* (%)	44 (46.8)	62 (54.9)	0.248
ARB, *n* (%)	7 (7.5)	10 (8.6)	0.714
ACEI, *n* (%)	44 (46.8)	54 (46.6)	0.982
Aspirin, *n* (%)	62 (66.0)	82 (70.7)	0.303
1 year medical treatment			
Beta-blockers, *n* (%)	76 (80.8)	98 (84.5)	0.543
CCBs, *n* (%)	26 (27.7)	25 (21.6)	0.283
Statins, *n* (%)	84 (89.4)	107 (92.2)	0.623
ARB, *n* (%)	62 (65.6)	76 (65.5)	0.861
ACEI, *n* (%)	7 (7.5)	11 (9.5)	0.616
Aspirin, *n* (%)	80 (85.1)	102 (87.9)	0.682
10 years medical treatment			
Beta-blockers, *n* (%)	57 (60.6)	65 (56.0)	0.501
CCBs, *n* (%)	23 (24.5)	22 (19.0)	0.333
Statins, *n* (%)	62 (65.9)	69 (59.5)	0.335
ARB, *n* (%)	41 (43.6)	51 (44.0)	0.959
ACEI, *n* (%)	19 (20.2)	14 (12.1)	0.106
Aspirin, *n* (%)	60 (63.8)	65 (56.0)	0.252

Notes: ACEI—angiotensin-converting enzyme inhibitor; ARB—angiotensin receptor blocker; CCB—calcium channel blocker.

**Table 6 biomedicines-12-01018-t006:** Results of binary logistic regression (forward LR method): association of factors with the risk of unfavorable long-term prognosis development after CABG.

		B	S.E.	Wald	df	Sig.	Exp(B)
All-cause mortality							
Step 1	Number of bypasses	0.551	0.213	6.704	1	0.010	1.735
	Constant	−2.646	0.593	19.912	1	0.000	0.071
Step 2	Group with worsened CAVI	0.876	0.376	5.421	1	0.020	2.400
	Number of bypasses	0.609	0.221	7.580	1	0.006	1.839
	Constant	−4.185	0.929	20.306	1	0.000	0.015
Step 3	Group with worsened CAVI	0.852	0.378	5.083	1	0.024	2.345
	Carotid stenoses ≥ 30%	0.843	0.432	3.810	1	0.051	2.323
	Number of bypasses	0.627	0.226	7.685	1	0.006	1.873
	Constant	−4.367	0.942	21.518	1	0.000	0.013
Combined endpoint (all-cause death + non-fatal myocardial infarction + non-fatal stroke)							
Step 1	Carotid stenoses ≥ 30%	1.186	0.389	9.293	1	0.002	3.274
	Constant	−0.829	0.173	22.982	1	0.000	0.436
Step 2	Group with worsened CAVI	0.853	0.322	6.995	1	0.008	2.346
	Carotid stenoses ≥ 30%	1.208	0.398	9.201	1	0.002	3.347
	Constant	−2.166	0.549	15.595	1	0.000	0.115

**Table 7 biomedicines-12-01018-t007:** Receiver operating characteristic curve analysis. Performance of baseline parameters (values of CAVI and presence of carotid stenoses ≥ 30%) in discriminating combined endpoint development in ten-year period after CABG. Area under the curve.

Test Result Variable(s)				Asymptotic 95% Confidence Interval	
	Area	Std. Error	Asymptotic Sig.	Lower Bound	Upper Bound
R_CAVI baseline	0.541	0.043	0.326	0.457	0.626
L_CAVI baseline	0.530	0.043	0.481	0.446	0.613
R_CAVI one year follow up	0.550	0.042	0.235	0.467	0.633
L_CAVI one year follow up	0.532	0.041	0.448	0.452	0.612
Group with worsened CAVI	0.596	0.041	0.023	0.516	0.676
Carotid stenoses ≥ 30%	0.577	0.043	0.068	0.493	0.661

**Table 8 biomedicines-12-01018-t008:** Receiver operating characteristic curve analysis. Performance of baseline parameters (values of CAVI and presence of carotid stenoses ≥ 30%) in discriminating all-cause mortality development in ten-year period after CABG. Area under the curve.

Test Result Variable(s)				Asymptotic 95% Confidence Interval	
	Area	Std. Error	Asymptotic Sig.	Lower Bound	Upper Bound
R_CAVI baseline	0.541	0.049	0.397	0.444	0.637
L_CAVI baseline	0.514	0.049	0.767	0.419	0.610
R_CAVI one year follow up	0.567	0.048	0.165	0.472	0.662
L_CAVI one year follow up	0.547	0.047	0.330	0.455	0.639
Group with worsened CAVI	0.590	0.046	0.062	0.499	0.681
Carotid stenoses ≥ 30%	0.554	0.050	0.266	0.456	0.651

## Data Availability

The datasets used and/or analyzed during the current study are available from the corresponding author on reasonable request.

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
