# Peer review of "Dynamics of the State of Arterial Stiffness as a Possible Pathophysiological Factor of Unfavorable Long-Term Prognosis in Patients after Coronary Artery Bypass Grafting"

_biomedicines, 2024, doi:10.3390/biomedicines12051018_

Round 1

Reviewer 1 Report

Comments and Suggestions for Authors

To the authors, 

 In this retrospective, single-center and observational study, Sumin AN, et al assessed serial changes of CAVI in patients before and one year after CABG. They showed that worsened CAVI one year after CABG was associated with the incidence of all cause death and MACE (defined as all cause death, MI, stroke, and TIA) at 10-year follow up. They also showed that the worsened CAVI, number of shunts (probably the number of bypass graft), and the presence of carotid stenosis more than 30% were independent predictors of all cause death. Detailed analysis of very long-term follow-up data including CAVI in patients with CABG is valuable to present. However, there are several concerns which need to be addressed appropriately.

Major comments:

i.           The authors need to describe the detailed definitions of “improved CAVI” and “worsened CAVI”, including the range and how to treat CAVI values of right and left side for group assignment. Were there any cases with discordant of right and left side in the serial change of CAVI? 

ii.           The study flow chart (Figure 1) needs appropriate revision. In 2nd step, n=356 (immediate results) was pulled out from n=545. Therefore, number of excluded patients should be 189. The total number of excluded cases needs to be addressed to avoid confusion. In the column of “1 years”, 1 years should be “1 year”. “linical” should be “clinical”. In addition, the word font needs to be consistent (i.e. please keep in Times New Roman). Please keep consistency for capitalization of the head words. 

iii.           The terminology of “worsened CAVI” need to be kept through the manuscript. In some points “worsened CAVI” has been changed into “impaired CAVI” (figures and tables) or “negative dynamics of CAVI”. Very confusing. Especially, the positive and negative dynamics of CAVI are very confusing terms (readers cannot imagine how the CAVI value changed in positive and negative dynamics). Please fix them.  

iv.           The authors need to conduct survival time analysis (Kaplan-Meier analysis) for exploring the detailed association between the change of CAVI and clinical endpoints. 

v.           (Page 8, line 188 and 194); The term “death” should be changed to ”cardiac death” or “all cause death”. Could you show me the details of all cause death? If the considerable number or malignancies were involved, the authors should address the information in patient characteristics. 

vi.           The authors have already reported the importance of CAVI for long-term follow up in patients underwent CABG (ref #18; Sumin AN, et al. J Clin Med. 2022,11,4585.). The current study data must come from the same study cohort. The details of difference between two studies (JCM paper and the current study) need to be discussed in depth. In the “Introduction” section, the information of prior JCM paper need to be addressed briefly.

Minor comments: 

i.           (Page 2, line 76); “356 people” needs to be changed into “356 patients”. 

ii.           (Page 2, line 91); “low-density lipoprotein (LDL)” should be “low-density lipoprotein cholesterol (LDL-C)”.

iii.           (Page 9, line 206 and 211) “Table 5” should be “Table 6”.

iv.           (Page 9, line 215) “Table 6” should be “Table 7”.

v.           (Page 9, line 217) “Table 7” should be “Table 8”.

vi.           (Table 2 and 3); “basal” should be “baseline”, and “after year” should be “one year follow up”. The functional class was abbreviated as “FK” but “FC” might be better. 

vii.           (Table 2); FBS of group with impaired CAVI was described as “5.6 [5.2; ]”. Where is the number of 75 percentile? 

viii.           In Table 2, it seems hard to recognize the comparison of the data between baseline and one year follow up. I recommend creating the additional columns for one year follow up data.

Author Response

Reply to Reviewer 1

To the authors, 

 In this retrospective, single-center and observational study, Sumin AN, et al assessed serial changes of CAVI in patients before and one year after CABG. They showed that worsened CAVI one year after CABG was associated with the incidence of all cause death and MACE (defined as all cause death, MI, stroke, and TIA) at 10-year follow up. They also showed that the worsened CAVI, number of shunts (probably the number of bypass graft), and the presence of carotid stenosis more than 30% were independent predictors of all cause death. Detailed analysis of very long-term follow-up data including CAVI in patients with CABG is valuable to present. However, there are several concerns which need to be addressed appropriately.

Reply: We are grateful to the reviewer for the work done in reviewing our manuscript and for useful comments that will help improve our article.

Major comments:

  1. The authors need to describe the detailed definitions of “improved CAVI” and “worsened CAVI”, including the range and how to treat CAVI values of right and left side for group assignment. Were there any cases with discordant of right and left side in the serial change of CAVI? 

Reply: We have clarified the text of the manuscript in section 2.1. Study Population. Now the text of this section is as follows:

One year after surgery, 251 patients (71%) were able to remeasure CAVI, and we tracked changes and classified them as having improved CAVI or worsened CAVI. The improved CAVI group included patients with a decrease in CAVI from a pathological value (CAVI≥9.0) to normal (CAVI<9.0) or the index remained within normal values. The worsened CAVI group included patients with a persistent pathological index value (CAVI≥9.0) or an increase from CAVI<9.0 to CAVI≥9.0 or an increase of 1 unit or more). If the CAVI value was ≥9.0 on at least one side, the index was considered pathological.

  1. The study flow chart (Figure 1) needs appropriate revision. In 2ndstep, n=356 (immediate results) was pulled out from n=545. Therefore, number of excluded patients should be 189. The total number of excluded cases needs to be addressed to avoid confusion. In the column of “1 years”, 1 years should be “1 year”. “linical” should be “clinical”. In addition, the word font needs to be consistent (i.e. please keep in Times New Roman). Please keep consistency for capitalization of the head words. 

Reply: We have made the necessary changes to the study flowchart (Figure 1).

iii.           The terminology of “worsened CAVI” need to be kept through the manuscript. In some points “worsened CAVI” has been changed into “impaired CAVI” (figures and tables) or “negative dynamics of CAVI”. Very confusing. Especially, the positive and negative dynamics of CAVI are very confusing terms (readers cannot imagine how the CAVI value changed in positive and negative dynamics). Please fix them.  

Reply: We have made the necessary corrections to the text of the manuscript.

  1. The authors need to conduct survival time analysis (Kaplan-Meier analysis) for exploring the detailed association between the change of CAVI and clinical endpoints. 

Reply: We additionally performed survival time analysis (Kaplan-Meier analysis with log runk tests) to examine the detailed association between change in CAVI and clinical endpoints. We have included relevant information in the text of the manuscript.

  1. (Page 8, line 188 and 194); The term “death” should be changed to ”cardiac death” or “all cause death”. Could you show me the details of all cause death? If the considerable number or malignancies were involved, the authors should address the information in patient characteristics. 

Reply: We have added information on causes of death to Supplementary Table S1:

Group with improved CAVI (n=94)

Group with worsened

CAVI (n=116)

P-value

All cause death

14(15,8)

32(27,6)

0,029

Death from cardiac causes

9(9,6)

17(14,7)

0,266

Death from non-cardiac causes

5(5,3)

14(12,0)

0,089

Death of malignancies

2(2,1)

9(7,8)

0,07

Covid 2019

1(1,1)

3(2,6)

0,422

The cause of death is unknown

0 (0)

1(0,9)

0,367

  1. The authors have already reported the importance of CAVI for long-term follow up in patients underwent CABG (ref #18; Sumin AN, et al. J Clin Med. 2022,11,4585.). The current study data must come from the same study cohort. The details of difference between two studies (JCM paper and the current study) need to be discussed in depth. In the “Introduction” section, the information of prior JCM paper need to be addressed briefly.

Reply: We have added information about our previous research in the Introduction section:

In our previous study [15], we showed that coronary artery disease patients with abnormal CAVI before CABG experienced more frequent side effects and death during long-term follow-up than patients with normal CAVI. It was also found that the presence of subclinical multifocal atherosclerosis and pathological CAVI were independent predictors of the development of the combined endpoint. However, the possible influence of the dynamics of CAVI on the long-term prognosis in this category of patients remains unclear.

Minor comments: 

  1. (Page 2, line 76); “356 people” needs to be changed into “356 patients”. 
  2. (Page 2, line 91); “low-density lipoprotein (LDL)” should be “low-density lipoprotein cholesterol (LDL-C)”.

iii.           (Page 9, line 206 and 211) “Table 5” should be “Table 6”.

  1. (Page 9, line 215) “Table 6” should be “Table 7”.
  2. (Page 9, line 217) “Table 7” should be “Table 8”.
  3. (Table 2 and 3); “basal” should be “baseline”, and “after year” should be “one year follow up”. The functional class was abbreviated as “FK” but “FC” might be better. 

vii.           (Table 2); FBS of group with impaired CAVI was described as “5.6 [5.2; ]”. Where is the number of 75 percentile? 

viii.           In Table 2, it seems hard to recognize the comparison of the data between baseline and one year follow up. I recommend creating the additional columns for one year follow up data.

Reply: We have made the all necessary corrections to the text of the manuscript.

Reviewer 2 Report

Comments and Suggestions for Authors

General comment: the drop-out rate is worryingly high. It could affect the findings. 

Table 2 &7

For each of the parameters: after year --> afte one year

Table 3

Terminology should be two and three vessel disease 

Table 5

Use medical treatment instead of medicine therapy

Lines 206 & 211: I assume this is about table 6, not table 5. 

Table 6 & line 204: shunt = bypass? 

The AUC in the ROC are low, this should be stated in the section limitations 

Comments on the Quality of English Language

There are only minor language issues 

Author Response

Reply to Reviewer 2

General comment: the drop-out rate is worryingly high. It could affect the findings. 

Reply: We are grateful to the reviewer for the work done in reviewing our manuscript and for useful comments that will help improve our article. In the flowchart we presented general data on the inclusion of patients and the formation of the cohort examined. Among the patients from whom information on the annual dynamics of CAVI was obtained (251 patients), 10-year follow-up information was obtained for 84% of patients, which is a fairly high result for such a long observation period.

Table 2 &7

For each of the parameters: after year --> afte one year

Reply: We have made the necessary corrections to the text of the manuscript.

Table 3

Terminology should be two and three vessel disease 

Reply: We have made the necessary corrections to the text of the manuscript.

Table 5

Use medical treatment instead of medicine therapy

Reply: We have made the necessary corrections to the text of the manuscript.

Lines 206 & 211: I assume this is about table 6, not table 5. 

Reply: We have made the necessary corrections to the text of the manuscript.

Table 6 & line 204: shunt = bypass? 

Reply: We have made the necessary corrections to the text of the manuscript.

The AUC in the ROC are low, this should be stated in the section limitations 

Reply: We have made the necessary corrections to the text of the manuscript and added to Study limitation:It should also be noted that the AUC in the ROC are low, indicating an unacceptable ability the studied factors to distinguish the development of adverse outcomes”

Round 2

Reviewer 1 Report

Comments and Suggestions for Authors

To the authors, 

The quality of the manuscript has been improved according to the authors’ revision. Several minor points still exist to be fixed appropriately. See the following comments. 

1.     The authors added two new figures (new Figure 3 and 4; Kaplan-Meier analyses). Accordingly, the figure numbers of ROC analyses should be “Figure 5 and 6”. Please fix them. Also, in the main document, figure numbers (Page 11, line 247-249) should be changed appropriately.

2.     (Table 5) “1 years” should be “1 year”.

3.     (Figure 2) The color of bar graph for “improved CAVI group” and “worsened CAVI group” are not consistent within the figure. Please keep consistency. 

4.     Please carefully look through the manuscript again to fix minor mistakes and typos.

Author Response

The quality of the manuscript has been improved according to the authors’ revision. Several minor points still exist to be fixed appropriately. See the following comments. 

  1. The authors added two new figures (new Figure 3 and 4; Kaplan-Meier analyses). Accordingly, the figure numbers of ROC analyses should be “Figure 5 and 6”. Please fix them. Also, in the main document, figure numbers (Page 11, line 247-249) should be changed appropriately.
  2. (Table 5) “1 years” should be “1 year”.
  3. (Figure 2) The color of bar graph for “improved CAVI group” and “worsened CAVI group” are not consistent within the figure. Please keep consistency. 
  4. Please carefully look through the manuscript again to fix minor mistakes and typos.

Reply: We are grateful to the reviewers for their high assessment of the work we have done and for further clarifying comments. We have made changes to the text of the manuscript; they are highlighted with a marker of a different color.

Reviewer 2 Report

Comments and Suggestions for Authors

some minor language must be addressed (ex: line 186: arterial basin = myocardial aera of tissue depending on coronary artery? 

line 238 should refer to table 6 

Comments on the Quality of English Language

some minor language must be addressed (ex: line 186: arterial basin = myocardial aera of tissue depending on coronary artery? 

Author Response

some minor language must be addressed (ex: line 186: arterial basin = myocardial aera of tissue depending on coronary artery? 

line 238 should refer to table 6 

Reply: We are grateful to the reviewers for their high assessment of the work we have done and for further clarifying comments. We have made changes to the text of the manuscript; they are highlighted with a marker of a different color.
